# Targeting Glutamine Transporters as a Novel Drug Therapy for Synovial Sarcoma

**DOI:** 10.3390/cancers18010015

**Published:** 2025-12-19

**Authors:** Tran Duc Thanh, Naoki Takada, Hana Yao, Yoshitaka Ban, Naoto Oebisu, Manabu Hoshi, Nguyen Tran Quang Sang, Nguyen Van Khanh, Dang Minh Quang, Le Thi Thanh Thuy, Tran Trung Dung, Hidetomi Terai

**Affiliations:** 1Department of Orthopedic Surgery, Osaka Metropolitan University Graduate School of Medicine, Osaka 545-8585, Japan; v.thanhtd26@vinmec.com (T.D.T.);; 2Sarcoma Center, Vinmec Times City International Hospital, Vinmec Healthcare System, Hanoi 11622, Vietnam; 3Department of Orthopedic Surgery, College of Health Science, VinUniversity, Hanoi 12400, Vietnam; 4Department of Orthopedic Surgery, Osaka City Juso Hospital, Osaka 532-0034, Japan; 5Department of Orthopedic Surgery, Osaka City General Hospital, Osaka 534-0021, Japan; 6Department of Global Education and Medical Sciences, Osaka Metropolitan University Graduate School of Medicine, Osaka 545-8585, Japan

**Keywords:** synovial sarcoma, glutamine metabolism, ASCT2, V9302, AKT/mTOR pathway

## Abstract

Synovial sarcoma is an aggressive cancer with limited treatment options. This study found that synovial sarcoma cells rely heavily on glutamine for growth. The glutamine transporter ASCT2 was highly expressed in patient samples. Blocking ASCT2 with the inhibitor V9302 reduced cancer cell growth, triggered apoptosis, and suppressed the AKT/mTOR signaling pathway in both in vitro and in vivo models, without causing serious side effects. These findings suggest that targeting glutamine metabolism through ASCT2 inhibition could be a promising new treatment strategy for synovial sarcoma.

## 1. Introduction

SS is a rare, aggressive, and high-grade mesenchymal tumor that accounts for 5–10% of soft tissue sarcomas and predominantly affects the extremities of young adults [1,2]. Despite intensive multimodal therapies—including radical surgery, radiotherapy, and chemotherapy—SS is frequently associated with a relatively poor prognosis. While the 5-year overall survival rate for children and adolescents with localized disease is favorable, reported at approximately 80–90% in large international cohorts [3], the prognosis remains poor for patients with metastatic or recurrent disease. About 40–50% of adult cases eventually develop metastases, where long-term survival rates drop around 60% [3,4,5]. The limited efficacy of existing treatments underscores the urgent need for novel therapeutic strategies.

In recent years, tumor cell metabolomics has garnered increasing attention, with the dependency on glutamine, known as “glutamine addiction,” emerging as a promising anti-cancer approach [6,7,8]. This dependency has been established as a therapeutic target in numerous malignancies, including several sarcoma subtypes such as Ewing sarcoma, chondrosarcoma, rhabdomyosarcoma, undifferentiated pleomorphic sarcoma (UPS), fibrosarcoma, and leiomyosarcoma [9,10,11,12]. However, the role of glutamine metabolism in SS remains to be established.

ASCT2 is a high-affinity, sodium-dependent, neutral amino acid transporter responsible for glutamine uptake in diverse cancer cells [13]. Overexpression of ASCT2 in various cancers, including melanoma, breast, gastric, and colorectal cancers, has been strongly associated with increased proliferation and poor prognosis, making it an attractive therapeutic target [6,14]. Moreover, V9302, a potent and selective ASCT2 inhibitor, has demonstrated preclinical anti-tumor efficacy in various cancers [15,16,17,18,19]; however, its role in SS remains unclear. Therefore, the purpose of this study was to functionally validate the glutamine dependency of SS and assess the therapeutic potential of targeting ASCT2 with V9302, both in vitro and in vivo.

## 2. Materials and Methods

### 2.1. Cell Culture

HS-SY-II and HEK293 cells were purchased from RIKEN BioResource Research Center, Kyoto, Japan [20]. Both cell lines were maintained in Dulbecco’s Modified Eagle’s medium (DMEM; Gibco, New York, NY, USA) supplemented with 10% fetal bovine serum (FBS; Gibco) and 1% penicillin-streptomycin (Wako, Osaka, Japan). All cells were cultured at 37 °C under humidified conditions with 5% CO_2_.

Primary SS tumor tissues were obtained immediately after surgery, promptly minced with sterile scissors, and digested with collagenase II (Worthington, Lakewood, NJ, USA) for 15 min. The suspension was then mechanically dissociated using an 18-gauge needle. and filtered through a 70-μm nylon mesh (BD Falcon, Franklin Lakes, NJ, USA) to obtain tumor cells. The isolated patient-derived primary SS cells (SSH1) were maintained in DMEM supplemented with 10% FBS and 1% penicillin-streptomycin in a 10 mm dish at 37 °C with a humidified atmosphere containing 5% CO_2_. For all experiments, cells were used at passage three or four.

### 2.2. CCK8 Assay

HS-SY-II, SSH1, and HEK293 cells were seeded in 96-well plates (1 × 10^4^ cells/well) and incubated for 24 h for attachment. To assess glutamine dependency, the medium was replaced with glutamine-free Advanced MEM (Gibco) containing different concentrations of glutamine for 4 days. To determine the effect of V9302 (TargetMol, Boston, MA, USA), the cells were treated with increasing concentrations for 3 days.

Cell viability was quantified using a Cell Counting Kit-8 (CCK-8, Dojindo Laboratories, Kumamoto, Japan) according to the manufacturer’s protocol. The absorbance was measured at 450 nm using a Varioskan LUX microplate reader (Thermo Fisher Scientific, Waltham, MA, USA). The 50% inhibitory concentration (IC_50_) values were calculated using GraphPad Prism 8.4.3 (GraphPad Software, Inc., La Jolla, CA, USA).

### 2.3. Apoptosis Assay

In brief, HS-SY-II and HEK293 cells were seeded in 6-well plates (0.5 × 10^6^ cells/well) and treated with V9302 (0, 15, 20, and 25 µM) for 24 h. Following treatment, both floating and adherent cells were harvested and stained using the FITC Annexin V Apoptosis Detection Kit I (BD Biosciences, San Jose, CA, USA) according to the manufacturer’s protocol. Samples were immediately analyzed using a BD LSRFortessa X-20 flow cytometer (BD Biosciences), and the total apoptotic population was determined by summing the early and late apoptotic rates.

### 2.4. Western Blotting Analysis

Briefly, HS-SY-II and HEK293 cells were cultured in increasing glutamine concentrations or treated with V9302 (0, 15, and 20 µM) for 24 h and lysed in RIPA buffer (Wako, Osaka, Japan) containing protease and phosphatase inhibitors (Roche Applied Science, Madison, WI, USA and Thermo Fisher Scientific, respectively). The protein concentration was determined using a BCA Protein Assay Kit (Thermo Fisher Scientific). Equal amounts of proteins were separated by sodium dodecyl sulfate-polyacrylamide gel electrophoresis (Bio-Rad, Hercules, CA, USA), transferred to PVDF membranes (Merck Millipore, Burlington, MA, USA), and blocked with 5% non-fat milk in Tris-buffered saline containing Tween-20 (TBST, Bio-Rad). Membranes underwent overnight incubation with primary antibodies at 4 °C (Table 1). After washing with TBST, the samples were incubated with appropriate horseradish peroxidase-conjugated secondary antibodies (Dako, Santa Clara, CA, USA). Immunoreactive bands were detected using an enhanced chemiluminescence substrate (ImmunoStar LD, Wako), with GAPDH as the loading control. Quantitative analysis was carried out with the ImageJ software (version 1.54 g, National Institutes of Health, Bethesda, MD, USA).

### 2.5. In Vivo Experiment

Four-week-old female BALB/c-nu/nu mice (Japan SLC, Inc., Tokyo, Japan) were subcutaneously injected in the right flank with 3 × 10^6^ HS-SY-II cells. When the tumors reached approximately 100 mm^3^, the mice were randomized (*n* = 5 per group) to receive daily intraperitoneal injections of either vehicle (phosphate-buffered saline with 5% dimethyl sulfoxide [DMSO]) or V9302 with 5% DMSO (30 mg/kg). Body weight and tumor volume [(length × width^2^)/2] were monitored every three days. After 21 days of treatment, the mice were euthanized, and the tumors, liver, kidney tissue, and plasma were harvested for analysis.

### 2.6. Immunohistochemical Analysis

IHC as conducted on formalin-fixed, paraffin-embedded sections of patient tumors (SS, *n* = 3; LPS, *n* = 3; Table 2) and mouse xenografts. Following deparaffinization and rehydration, antigen retrieval was carried out by heating in a water bath with Target Retrieval Solution (Dako). After blocking steps, sections were incubated overnight at 4 °C with primary antibodies against ASCT2, p-S6 (Phospho-S6 ribosomal protein), or cleaved caspase-3 (Table 1). The next day, sections were incubated with the appropriate horseradish peroxidase-conjugated secondary antibody for 1 h at room temperature. The immunoreactive signal was visualized using a DAB (3,3′-Diaminobenzidine) substrate kit (Nichirei Biosciences, Tokyo, Japan), followed by hematoxylin counterstaining. Slides were imaged using a BZ-X700 microscope (KEYENCE, Osaka, Japan), and the percentage of DAB-positive area was quantified from ten 400× fields per section using ImageJ.

### 2.7. Statistical Analysis

Data analyses were performed using GraphPad Prism version 8.4.3. Data were analyzed using the student’s *t*-test, one- or two-way ANOVA, linear trend test, linear mixed-effects models, or nonlinear regression, as appropriate. Statistical significance was defined as *p* < 0.05.

## 3. Results

### 3.1. ASCT2 Is Highly Expressed and Constitutive in SS

In SS, *SLC1A5*, which encodes the glutamine transporter ASCT2, was identified as one of the most highly expressed glutamine transporter-related genes by single-cell RNA sequencing (Appendix A) [21]. To validate this observation, IHC was performed to compare ASCT2 protein levels in surgical specimens from patients with SS and LPS. The results revealed that SS tissues exhibited robust and diffuse ASCT2 expression, whereas LPS and adjacent non-tumor tissues showed lower levels (Figure 1a). Quantitative analysis confirmed that the percentage of ASCT2-positive cells was significantly higher in SS tissues than in LPS and adjacent non-tumor tissues (*p* < 0.001) (Figure 1b).

To determine if ASCT2 expression is regulated by glutamine availability, we assessed ASCT2 protein levels under varying glutamine concentrations. Western blot analysis revealed that ASCT2 expression remained stable and did not change significantly in response to glutamine deprivation or supplementation (Figure 1c), suggesting that high ASCT2 expression in SS is constitutive.

### 3.2. SS Cells Exhibit High Glutamine Demand for Proliferation

We investigated the effects of glutamine on SS cell proliferation. When cultured in a glutamine-free medium, HS-SY-II cells demonstrated significant growth arrest over a four-day period, whereas control HEK293 cells were less affected (*p* < 0.001) (Figure 2a), indicating a strong dependence of HS-SY-II cells on glutamine. Glutamine supplementation supported sustained cell proliferation in a time-dependent manner, whereas cells in the glutamine-free medium failed to proliferate (Figure 2b). Moreover, the viability of both HS-SY-II and SSH1 (a primary tumor cell derived from a patient with SS) cells increased progressively with increasing glutamine concentrations from 0 to 4 mM (Figure 2c,d). Collectively, these findings indicate that glutamine is a critical nutrient for SS proliferation.

To determine the mechanism of HS-SY-II cell proliferation in the presence of glutamine, we assessed the phosphorylation status of key signaling proteins of AKT/mTOR pathway in HS-SY-II cells cultured under increasing glutamine concentrations (0, 0.25, 1, and 4 mM) for 24 h. As expected, phosphorylation of mTOR (p-mTOR) and its downstream effector S6 (p-S6) increased significantly with higher glutamine concentrations, confirming that mTOR activity is nutrient-responsive. In contrast, p-AKT levels exhibited an inverse trend, being highest under glutamine starvation (0 mM) and progressively decreasing as glutamine concentration increased (Appendix A). This observation suggests that glutamine depletion itself modulates AKT/mTOR signaling as a compensatory survival mechanism in SS cells.

### 3.3. V9302 Suppresses Cell Viability in SS While Sparing Normal Cells

In HS-SY-II cells, V9302 induced a dose-dependent reduction in cell viability, with an IC_50_ of 14.11 µM (Figure 3a). A similar dose-dependent decrease was observed in SSH1 cells (Figure 3b). Furthermore, the anti-proliferative effect of V9302 was time-dependent. While untreated cells continued to proliferate, treatment with 25 µM and 30 µM V9302 significantly reduced HS-SY-II cell viability after 1, 2, and 3 days (*p* < 0.05) (Figure 3c). Interestingly, V9302 exhibited high selectivity, effectively reducing the viability of HS-SY-II cells, while minimally affecting HEK293 cells across the same concentration range (Figure 3d).

### 3.4. V9302 Suppresses the AKT/mTOR Signaling Pathway

To elucidate the molecular mechanism responsible for the effects of V9302, we performed Western blotting analysis to assess the AKT/mTOR signaling pathway. Treatment of HS-SY-II with V9302 (15 and 20 µM) led to a dose-dependent reduction in the phosphorylation levels of AKT, mTOR, p70S6K, and S6 protein (Figure 4a). Quantitative analysis of Western blot data confirmed a significant dose-dependent reduction in the ratio of phosphorylated to total protein for each component (*p* < 0.05) (Figure 4b). While passive glutamine starvation allows cells to upregulate p-AKT to resist death (Appendix A), V9302 treatment results in the simultaneous inhibition of both p-mTOR and p-AKT (Figure 4a). This indicates that V9302 exerts a more potent blockade that overrides this compensatory survival loop, thereby acting more effectively than nutrient deprivation alone.

To further investigate the selectivity of V9302, we analyzed signaling pathways in normal HEK293 cells. Unlike SS cells, where V9302 suppressed both AKT and mTOR, V9302 treatment in HEK293 cells reduced p-S6 but paradoxically increased p-AKT levels (Appendix A). This suggests that normal cells engage a compensatory survival mechanism to resist metabolic stress, explaining their resistance to V9302 of HEK293 cells.

### 3.5. V9302 Induces Apoptosis in SS but Not in Normal Cells

To determine the mechanism of V9302-induced cell death, we performed an apoptosis assay using Annexin V and propidium iodide staining. Treatment of HS-SY-II cells with V9302 led to a significant, dose-dependent increase in the percentage of apoptotic cells, including early and late apoptotic cells, whereas HEK293 cells remained largely unaffected (*p* < 0.05) (Figure 5a,b).

Western blotting analysis was performed to determine the apoptotic mechanism of V9302. Treatment of HS-SY-II cells with V9302 at 15 µM and 20 µM resulted in a dose-dependent increase in the level of cleaved caspase-3, the active form of a key executioner caspase [22]. Quantitative analysis confirmed that V9302 treatment significantly upregulated the ratio of cleaved caspase-3 to total caspase-3 compared with that in the control group (*p* < 0.05) (Figure 5c,d). Collectively, these findings indicate that V9302 induces apoptosis in HS-SY-II cells by activating the caspase-3 pathway.

### 3.6. V9302 Inhibits Tumor Growth in the SS Xenograft Model by Modulating mTOR and Apoptosis Signaling

The in vivo anti-tumor efficacy of V9302 was evaluated using a mouse xenograft model. Treatment with V9302 markedly reduced the tumor size compared with the control treatment (Figure 6a,b). Furthermore, tumor weight was significantly lower in the V9302-treated group than in the control group (*p* < 0.001) (Figure 6c).

To confirm that the in vivo anti-tumor activity was driven by the same mechanisms observed in vitro, harvested xenograft tumors were subjected to IHC. Tumors from the V9302-treated group exhibited a significant reduction in p-S6 staining, confirming inhibition of the mTOR pathway in vivo. Moreover, these tumors showed a significant increase in cleaved caspase-3 staining, indicating that V9302 induced apoptosis in SS in vivo (*p* < 0.01) (Figure 6d,e).

V9302 was well-tolerated in experimental animals, suggesting minimal systemic toxicity. No significant differences in body weight were observed between the V9302-treated and control groups throughout the study period (Figure 7a). Additionally, hematoxylin-eosin (H&E) staining of liver and kidney tissues revealed no apparent pathological changes or signs of organ damage in V9302-treated group compared with the control group (Figure 7b). Importantly, V9302 treatment did not significantly alter liver or kidney function markers (alanine transaminase, aspartate transaminase, and creatinine) compared with those in the control group (Figure 7c–e).

## 4. Discussion

In this work, we examined the glutamine dependence of SS and the effects of the ASCT2 inhibitor V9302, both in vitro and in vivo. Our IHC analysis of patient-derived SS tumor specimens revealed high levels of ASCT2 protein expression compared with those in LPS and non-tumor tissues, providing a direct clinical correlation and a potential target for therapeutic intervention. This finding is consistent with previous studies reporting markedly elevated ASCT2 expression in non-small-cell lung, liver, gastric, and colon cancers [14], representing a common metabolic alteration of glutamine in multiple cancer types. Analysis of protein levels corroborated previous single-cell RNA-sequencing data [21], which reported substantially increased *SLC1A5* mRNA expression—the gene encoding the ASCT2 protein—in SS. Furthermore, ASCT2 overexpression has been associated with poor survival in squamous cell carcinoma, gastric cancer, breast cancer, and ovarian cancer [8,14,23,24]. Importantly, we observed that ASCT2 levels in SS cells remained stable regardless extracellular glutamine concentration. This constitutive overexpression suggests that ASCT2 is intrinsically driven by oncogenic signaling in SS, ensuring that the target for V9302 remains present even within the fluctuating nutrient gradients of the tumor microenvironment.

This study, to our knowledge, is the first to demonstrate that SS exhibits a marked glutamine dependency for cell proliferation and survival. This contributed to a fundamental metabolic vulnerability in many other cancers, often termed “glutamine addiction” [5,8,25,26,27]. We further demonstrated glutamine dependence of SS proliferation using patient-derived SS cells. These results suggest that targeting the glutamine metabolic pathway may represent an effective therapeutic strategy for SS. Glutamine dependency has also been observed in several other sarcoma subtypes, including UPS, fibrosarcoma, Ewing sarcoma, chondrosarcoma, and rhabdomyosarcoma [9,10,11]. Interestingly, a recent study reported no association between ASCT2 expression and the response to the inhibitor V9302 [28], suggesting that the extent of glutamine dependence in SS may be a more critical determinant of drug efficacy.

V9302 was identified as a lead compound in a 2015 screening process and demonstrated a 100-fold increase in potency compared with gamma-L-glutamyl-p-nitroanilide [29]. Our study is the first to assess this inhibitor in SS, yielding compelling results. In vitro, V9302 induced a dose- and time-dependent decrease in the viability of HS-SY-II and patient-derived SS cells. Importantly, this cytotoxic effect was selective to cancer cells, while exerting a minimal effect on non-cancerous HEK293 cells. These promising in vitro findings robustly translate into in vivo efficacy. In a subcutaneous HS-SY-II xenograft model, daily intraperitoneal administration of V9302 led to a statistically significant suppression of tumor growth compared with the control treatment. This anti-tumor effect was achieved with minimal systemic toxicity, as evidenced by the stable body weight of the treated group during the study period, the absence of pathological changes in H&E-stained liver and kidney tissue sections, and normal liver and kidney function in the treated group.

While Lee et al. suggested that CB-839, a glutaminase inhibitor, is a promising target for the UPS [9], our study using V9302 represents a broader intervention than solely targeting glutaminase alone. It should be noted that glutamine plays various metabolic roles immediately upon cellular entry, some of which are independent of its conversion to glutamate by glutaminase, such as the biosynthesis of hexosamine, nucleotides, and asparagine in the cytoplasm, maintenance of redox homeostasis, and activation of key signaling pathways such as mTOR and ERK [6,30]. By preventing glutamine entry into cells through ASCT2 inhibition, all glutamine-dependent pathways are simultaneously disrupted. This broad blockade of glutamine utilization contributes to potent anti-tumor effects and may offer a superior strategy for overcoming potential metabolic escape routes [7,8,16]. To clarify the molecular mechanisms of the anti-tumor effects of V9302 in SS, we investigated its effect on key cellular signaling and survival pathways. Our results revealed that ASCT2 inhibition modulates the AKT/mTOR pathway, a master regulator of cell growth and proliferation. Western blot analysis demonstrated that the treatment of HS-SY-II cells with V9302 dose-dependently decreased the phosphorylation of AKT and mTOR, and their canonical downstream effectors, p70S6K and p-S6. This provides a direct mechanistic link between the blockade of glutamine uptake and the shutdown of a critical pro-growth signaling cascade. Suppression of the pro-survival mTOR pathway, coupled with the profound metabolic stress induced by glutamine starvation, culminates in the activation of programmed cell death [31]. In our study, flow cytometric analysis showed a dose-dependent elevation in the population of apoptotic cells following V9302 treatment. Western blot analysis demonstrated a marked increase in the levels of cleaved caspase-3, an enzyme involved in apoptosis. Collectively, these findings suggest that V9302-mediated inhibition of ASCT2 not only suppresses cell growth but also exerts cytotoxic effects on SS cells.

Intriguingly, our mechanistic data revealed that while passive glutamine deprivation reduced mTOR activity, it paradoxically increased p-AKT levels (Appendix A). This is likely due to the release of the S6-dependent negative feedback loop on IRS-1, a common survival mechanism in cancer cells during nutrient stress [32]. Importantly, V9302 treatment did not trigger this compensatory AKT activation; instead, it potently suppressed both the AKT and mTOR signaling axes (Figure 4). This indicates that V9302 provides a more potent blockade that overrides this compensatory survival loop, inhibits the growth of cancer cells, increasing apoptosis more effectively than nutrient deprivation alone.

The selective toxicity of V9302 is further supported by the differential signaling responses observed between cancer and normal cells. In HS-SY-II cells, V9302 caused the complete collapse of the AKT/mTOR axis. In contrast, normal HEK293 cells responded to V9302 by downregulating p-S6 but upregulating p-AKT (Appendix A). This compensatory AKT activation likely serves as a protective survival signal, preserving the viability of normal tissues during treatment.

These in vitro findings were corroborated in vivo, as IHC analysis of xenograft tumors from V9302-treated mice showed a marked decrease in p-S6 levels and an increase in cleaved caspase-3 compared with controls. This mechanism is well established in other cancers [14,16], where ASCT2 inhibition reduces cancer cell growth and proliferation while promoting cell death. Moreover, a recent study revealed that V9302 not only inhibits ASCT2 but also the amino acid transporters SNAT2 and LAT1 [28]. Our study had some limitations. First, our conclusions are based on a limited number of SS cell lines; however, we confirmed that V9302 is effective in vitro against tumor cells derived from patients with SS. Notably, the effectiveness of ASCT2 inhibitors also depends on the cancer subtype, as one study reported that ASCT2 inhibition was only effective in triple-negative basal-like breast cancer [33]. Future studies should validate these findings using a broader panel of SS cell lines and additional patient-derived models. Second, the use of immunodeficient mice affected the assessment of the interactions between ASCT2 inhibition and the host immune system. This is a notable limitation, as emerging evidence suggests that V9302 promotes anti-tumor immunity [6,34]. A critical next step involves evaluating V9302 in immunocompetent mouse models of SS, potentially in combination with checkpoint inhibitors.

## 5. Conclusions

In conclusion, this study presents the first direct functional evidence that SS is a glutamine-dependent malignancy reliant on high-level expression of the amino acid transporter ASCT2. Targeting this metabolic vulnerability with the potent and selective ASCT2 inhibitor V9302 effectively suppressed AKT/mTOR signaling, induced apoptosis, and inhibited tumor growth in preclinical models of SS. Collectively, these findings highlight a novel and highly effective therapeutic strategy for SS and highlight ASCT2 as a compelling target for the clinical development of ASCT2 inhibitors in patients with SS.

## Figures and Tables

**Figure 1 cancers-18-00015-f001:**
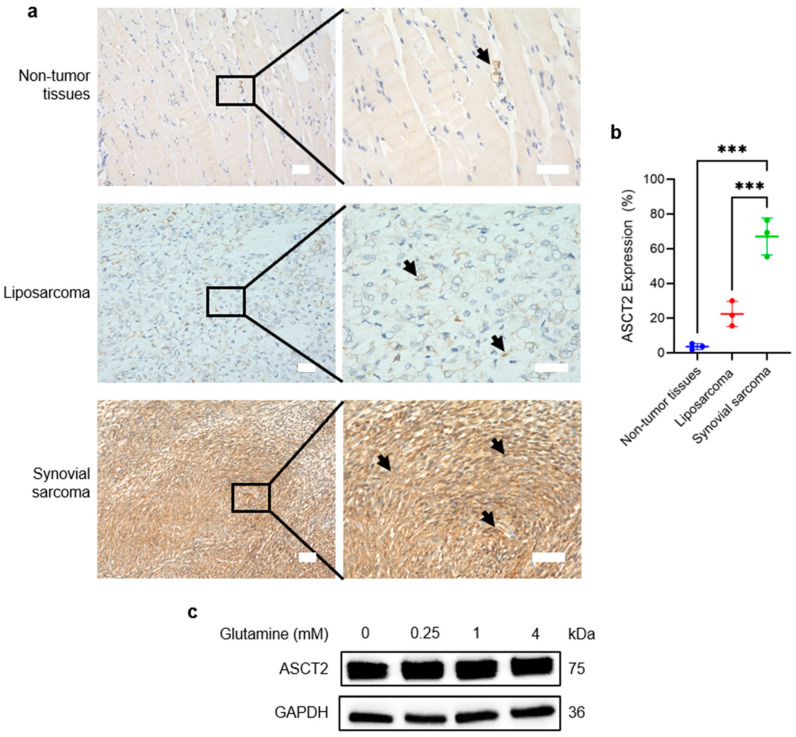
ASCT2 is highly expressed in synovial sarcoma tissues. (**a**) Representative immunohistochemistry images showing ASCT2 protein expression in non-tumor, liposarcoma (LPS), and synovial sarcoma (SS) tissues. Arrows indicate ASCT2-positive cells (brown staining). Scale bar, 50 µm. (**b**) Quantification of the percentage of ASCT2-positive cells from immunohistochemistry slides for non-tumor (*n* = 3), LPS (*n* = 3), and SS (*n* = 3) tissues. (**c**) Western blot analysis of ASCT2 in varying glutamine concentrations. Data are presented as mean ± SD. One-way analysis of variance (ANOVA) with Tukey’s test was used. *** *p* < 0.001. The uncropped bolts are shown in Appendix A.

**Figure 2 cancers-18-00015-f002:**
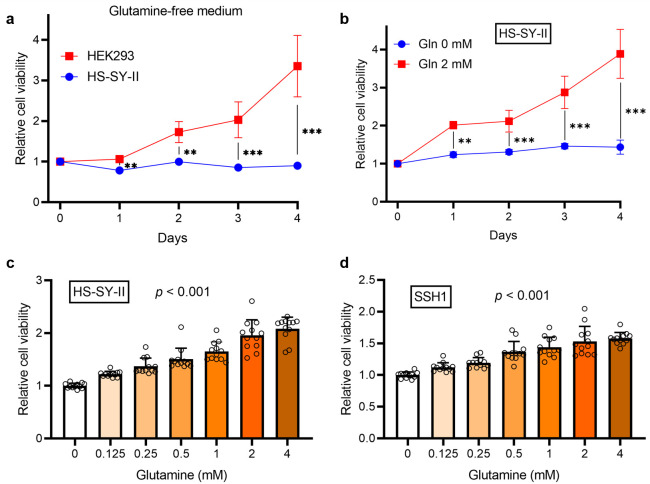
Synovial sarcoma exhibits strong glutamine dependency for proliferation. (**a**) Cell viability of HS-SY-II and HEK293 cells in glutamine-free medium over a 4-day period. (**b**) Proliferation of HS-SY-II cells over 4 days in medium with or without glutamine. (**c**) Proliferation of HS-SY-II cells cultured in medium containing increasing glutamine concentrations. (**d**) Proliferation of SSH1 (patient-derived synovial sarcoma cells) in increasing glutamine concentrations. Data are presented as mean ± SD and individual data points are shown as circles (*n* = 12). Two-way analysis of variance (ANOVA) with Sidak’s post hoc test was used in (**a**,**b**), ** *p* < 0.01; *** *p* < 0.001. A linear trend test was applied in (**c**,**d**).

**Figure 3 cancers-18-00015-f003:**
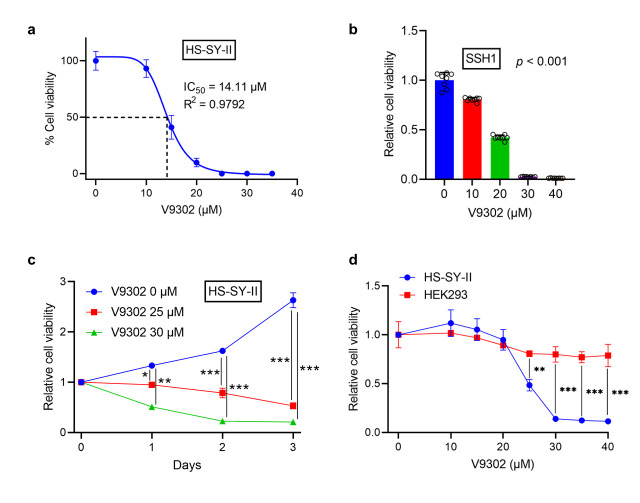
V9302 selectively reduces the viability of synovial sarcoma in a dose- and time-dependent manner. (**a**) Dose–response curve for HS-SY-II cells treated with V9302; IC_50_: half-maximal inhibitory concentration. (**b**) Viability of patient-derived synovial sarcoma cells (SSH1) treated with V9302. (**c**) Viability of HS-SY-II cells treated with V9302 for three days. (**d**) Viability of HS-SY-II and HEK293 cells treated with V9302. Data are presented as mean ± SD and individual data points are shown as circles (*n* = 8). A linear trend was applied in (**b**). Two-way analysis of variance (ANOVA) with Sidak’s multiple comparison test was used for (**c**,**d**). * *p* < 0.05; ** *p* < 0.01; *** *p* < 0.001.

**Figure 4 cancers-18-00015-f004:**
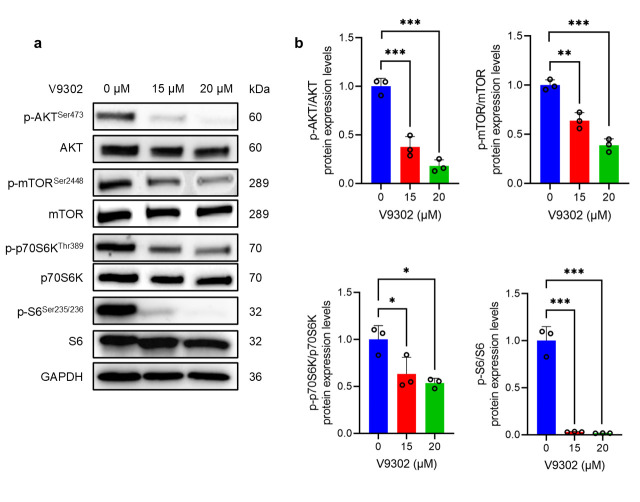
V9302 inhibits the AKT/mTOR signaling pathway. (**a**) Western blot analysis of key proteins in the AKT/mTOR pathway after treating HS-SY-II cells with V9302 (0, 15, 20 µM) for 24 h. GAPDH was used as a loading control. (**b**) Quantification of the Western blot bands. Data are presented as mean ± SD and individual data points are shown as circles (*n* = 3). One-way analysis of variance (ANOVA) with Tukey’s post hoc test was used. * *p* < 0.05; ** *p* < 0.01; *** *p* < 0.001. The uncropped bolts are shown in Appendix A.

**Figure 5 cancers-18-00015-f005:**
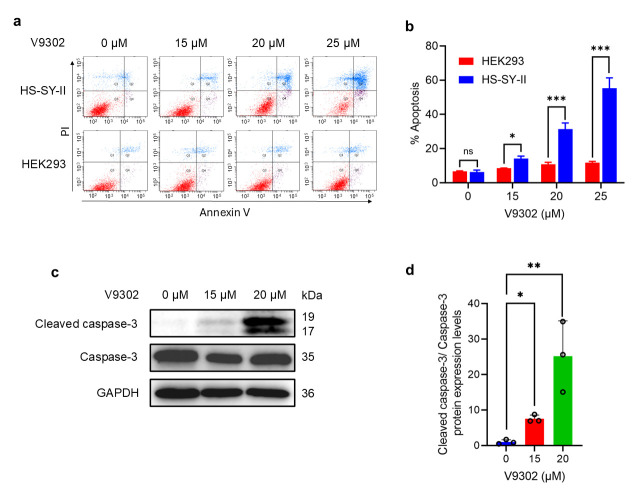
V9302 induces apoptosis in HS-SY-II but not in HEK293 cells. (**a**) Representative images of HS-SY-II and HEK293 cells treated with V9302 were subjected to apoptotic assays. (**b**) Quantification of the apoptotic rate of HS-SY-II and HEK293 cells using an apoptosis assay. (**c**) Western blot analysis showing induction of apoptosis by V9302. (**d**) Quantification of the cleaved caspase-3 protein levels. Data are presented as mean ± SD and individual data points are shown as circles (*n* = 3). Two-way analysis of variance (ANOVA) with Sidak’s multiple comparison test was used in (**b**). ANOVA with Dunnett’s test was used in (**d**). ns: not significant; * *p* < 0.05; ** *p* < 0.01; *** *p* < 0.001; ns: not significant. The uncropped bolts are shown in Appendix A.

**Figure 6 cancers-18-00015-f006:**
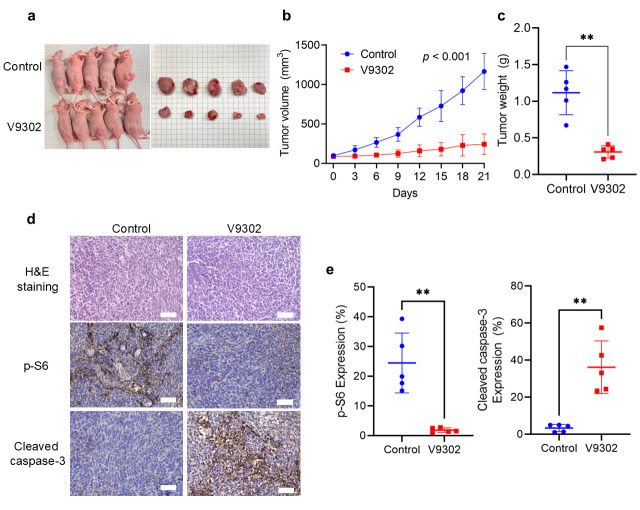
V9302 treatment suppresses tumor growth in a synovial sarcoma HS-SY-II xenograft model. (**a**) Representative images of mice and the corresponding excised tumors from the control and V9302-treated groups at the end of the experimental period (day 21). (**b**) Tumor volume of the control and V9302-treated groups over the 21-day treatment period. (**c**) Tumor weight of the control and V9302-treated groups at the end of the experimental period. (**d**) Hematoxylin-eosin (H&E) and immunohistochemical staining showing the expression of p-S6 and cleaved caspase-3 in tumor tissues of the control and V9302-treated group. Scale bar, 50 µm. (**e**) Quantification of p-S6 and cleaved caspase-3 in the control and V9302-treated groups. Data are presented as mean ± SD. A linear mixed-effects model was applied in (**b**). Student’s *t*-test was used in (**c**,**e**). ** *p* < 0.01.

**Figure 7 cancers-18-00015-f007:**
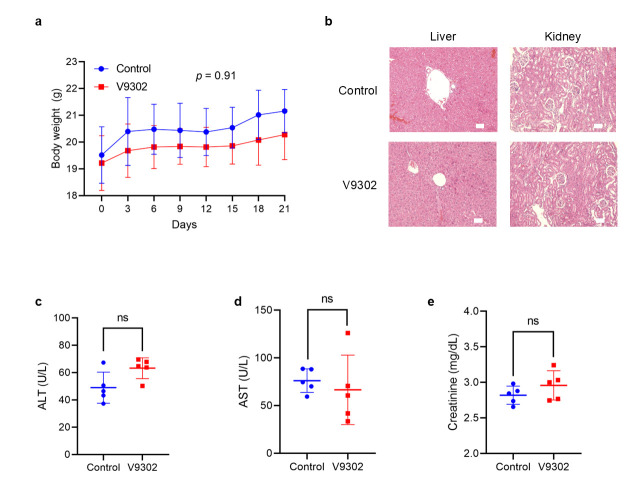
Side effects in the nontreated and V9302-treated groups. (**a**) Body weight of control and V9302-treated groups over the 21-day treatment period. (**b**) Hematoxylin-eosin staining of the liver and kidneys of control and V9302-treated groups. Scale bar, 50 µm. (**c**–**e**) Liver and kidney function markers in the control and V9302-treated groups. A linear mixed-effects model was applied in (**a**). Student’s *t*-test was used for (**c**,**e**) (ns: not significant). ALT, alanine transaminase; AST, aspartate transaminase.

**Table 1 cancers-18-00015-t001:** List of antibodies used in this study.

Antibody	Company	Dilution	Application
Anti-phospho-AKT	Cell Signaling (#9271)	1/1000	WB
Anti-AKT	Cell Signaling (#9272)	1/1000	WB
Anti-phospho-mTOR	Cell Signaling (5536T)	1/1000	WB
Anti-mTOR	Cell Signaling (#2972)	1/1000	WB
Anti-phospho-p70S6K	Cell Signaling (#9205)	1/1000	WB
Anti-p70S6K	Cell Signaling (#9202)	1/1000	WB
Anti-phospho-S6	Cell Signaling (#2211)	1/1000	WB, IHC
Anti-S6	Cell Signaling (2217T)	1/1000	WB
Anti-cleaved caspase-3	Cell Signaling (#9664)	1/1000	WB, IHC
Anti–caspase-3	Cell Signaling (#9662)	1/1000	WB
Anti-GAPDH	Millipore (MAB374)	1/10.000	WB
SLC1A5/ASCT2 Antibody	Proteintech (20350-1-AP)	1/400	WB, IHC

IHC, immunohistochemistry; WB, Western blotting.

**Table 2 cancers-18-00015-t002:** List of patients with SS and LPS.

Code	Age	Gender	Site	Pathology	Fusion Gene	Chemotherapy	Outcome
SSH1	28	Male	Thigh	Monophasic synovial sarcoma	SYT-SSX 1.2.4	Doxorubicin + Ifosfamide	CDF
SSH2	47	Male	Thigh	Biphasic synovial sarcoma	SYX-SST+	Doxorubicin + Ifosfamide	DOD
SSH3	52	Female	Foot	Monophasic synovial sarcoma	SS18+	Doxorubicin + Ifosfamide	CDF
LPS1	89	Male	Thigh	Dedifferentiated liposarcoma		No	DOD
LPS2	48	Male	Thigh	Myxoid liposarcoma		No	DOD
LPS3	63	Female	Thigh	Myxoid liposarcoma		No	CDF

CDF, continuous disease free; DOD, died of disease.

## Data Availability

The original contributions of this study are included in the article. Further inquiries can be directed to the corresponding author.

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
