# Peer review of "Targeting Glutamine Transporters as a Novel Drug Therapy for Synovial Sarcoma"

_cancers, 2025, doi:10.3390/cancers18010015_

Round 1

Reviewer 1 Report

Comments and Suggestions for Authors

The authors investigated glutamine dependency in SS and assessed the therapeutic potential of inhibiting the glutamine transporter ASCT2 using V9302. They concluded that targeting this metabolic vulnerability with the potent and selective ASCT2 inhibitor V9302 effectively suppressed AKT/mTOR signaling, induced apoptosis, and inhibited tumor growth in preclinical models of SS.

The topic is interesting because targeting metabolic pathways is a promising new approach in oncology. The manuscript is well-written and provides preliminary evidence that inhibiting glutamine transport in SS cells may represent a new therapeutic option.

Comments:

Abstract Synovial sarcoma (SS) is a malignant soft tissue neoplasm with aggressive clinical behavior, poor survival rates, and limited treatment options

That's not true. The treatment options are the same as for other soft tissue tumors: surgery, radiation therapy (RT), and chemotherapy. These treatments have produced very good results in adolescent andyoung adult patients with localized, resectable tumors.

ASCT2 expression was elevated in patients with SS.

Where exactly was it elevated?

Our findings establish SS as a glutamine-dependent malignancy and validate ASCT2 as a promising therapeutic target

Overinterpretation: The findings suggest this, but it has not been established.

Graphical abstract

Explain pS6

About 40% -50% of cases develop metastases, and the 5-year survival rates has been reported to be as low as 10% [3,4].

This is not true for younger patients. Please cite the results of international studies, such as EURAMUS. i.e. Eur J Cancer. 2019 Mar;109:36-50. doi: 10.1016/j.ejca.2018.11.027. Epub 2019 Jan 25.PMID: 30685685 

Taken together,these results suggest that V9302 effectively suppresses the AKT/mTOR signaling pathway in a dose-dependent manner.

Suppressing the AKT/mTOR signaling pathway does not necessarily mean that it is an effective anti-sarcoma (SS) compound, since many other AKT/mTOR inhibitors have been developed and used in oncology without producing meaningful results in sarcoma.

Author Response

Please kindly refer to the attached 'Response to Reviewers' file for our detailed point-by-point response

Reviewer 2 Report

Comments and Suggestions for Authors
  1. The manuscript does not clearly specify whether the single-cell RNA-seq dataset used for analysis is derived from Reference 20 or Reference 21. This should be explicitly stated to avoid confusion.
  2. The authors should provide baseline ASCT2 expression levels in all cell lines used in the study. This is essential for interpreting the results shown in Figure 3C. Additionally, it would be informative to examine whether ASCT2 expression changes dynamically in response to varying glutamine concentrations.
  3. The manuscript jumps to AKT/mTOR signaling analysis without establishing a mechanistic link. Since V9302 is described as an ASCT2 (glutamine transporter) inhibitor, the authors should first demonstrate whether glutamine depletion itself modulates AKT/mTOR signaling. This would provide a logical foundation for evaluating pathway responses to V9302.
  4. Because V9302 reportedly has no effect on HEK293 cells, the authors should also assess whether AKT/mTOR signaling remains unchanged in this cell line after V9302 treatment or glutamine starvation. This would strengthen the conclusion that HEK293 cells are functionally insensitive under these conditions.

Author Response

Please kindly refer to the attached 'Response to Reviewers' file for our detailed point-by-point response.
